# Design and In Situ Validation of Low-Cost and Easy to Apply Anti-Biofouling Techniques for Oceanographic Continuous Monitoring with Optical Instruments

**DOI:** 10.3390/s23020605

**Published:** 2023-01-05

**Authors:** Tiago Matos, Vânia Pinto, Paulo Sousa, Marcos Martins, Emilio Fernández, Renato Henriques, Luis Miguel Gonçalves

**Affiliations:** 1CMEMS-UMinho, Campus de Azurém, University of Minho, 4800-058 Guimarães, Portugal; 2LABBELS—Associate Laboratory, 4800-058 Guimarães, Portugal; 3INESC TEC, 4200-465 Porto, Portugal; 4Grupo de Oceanografía Biolóxica, Faculty of Marine Science, Universidade de Vigo, 36310 Vigo, Spain; 5Institute of Earth Sciences, Campus de Gualtar, University of Minho Pole, 4710-057 Braga, Portugal

**Keywords:** biofouling, optical sensors, transparent coating, biocide, chlorine, copper, oceanography

## Abstract

Biofouling is the major factor that limits long-term monitoring studies with automated optical instruments. Protection of the sensing areas, surfaces, and structural housing of the sensors must be considered to deliver reliable data without the need for cleaning or maintenance. In this work, we present the design and field validation of different techniques for biofouling protection based on different housing materials, biocides, and transparent coatings. Six optical turbidity probes were built using polylactic acid (PLA), acrylonitrile butadiene styrene (ABS), PLA with copper filament, ABS coated with PDMS, ABS coated with epoxy and ABS assembled with a system for in situ chlorine production. The probes were deployed in the sea for 48 days and their anti-biofouling efficiency was evaluated using the results of the field experiment, visual inspections, and calibration signal loss after the tests. The PLA and ABS were used as samplers without fouling protection. The probe with chlorine production outperformed the other techniques, providing reliable data during the in situ experiment. The copper probe had lower performance but still retarded the biological growth. The techniques based on transparent coatings, epoxy, and PDMS did not prevent biofilm formation and suffered mostly from micro-biofouling.

## 1. Introduction

Biofouling, or biological fouling, is the accumulation of biological material from molecules to metazoans on surfaces. Soon after a structure is immersed in water (fresh or salty) it is gradually covered by organisms which may compromise the operation of the devices. In some cases, this biofouling formation is virtually instantaneous [1]. These organisms can be divided according to their size, into primary film (biofouling) or micro-organisms (also called micro-fouling), and macro-fouling [2]. The succession of the fouling states can be divided into five main events [3]:Adsorption of organic and inorganic macromolecules immediately after immersion, forming the primary film.Transport of microbial cells to the surface and the immobilization of bacteria on the surface.Bacterial attachment to the substratum through extracellular polymer production, forming a microbial film on the surface.Development of a more complex community with the presence of multicellular species, microalgae, debris, sediments, etc., on the surface.Attachment of macroalgae and marine invertebrates such as barnacles or mussels.

However, the fouling formation is not standard, but a combination of different physical, chemical, and biological factors. The water properties (pH, salinity, temperature, conductivity, dissolved oxygen, etc.), hydraulic conditions, depth, season, and local fauna and flora species composition play a major role in the fouling development. Thus, while the formation of biofouling in the surfaces is certain, its development is very difficult to predict [4].

Biofouling is of extreme importance for monitoring instruments. Even a small amount of biofilm in the sensing area can produce undesired interferences in the readings of the sensor, which can no longer provide reliable information. These interferences can result from the clogging of sections, membranes, or filters in mechanical sensors (as also changes in the normal mechanical properties of MEMs sensors); contamination in chemical sensors (for example, macro-fouling in dissolved oxygen sensors); increasing resistance to heat exchange in thermal sensors; increasing of acoustic absorption and consequent decrease in receiving power in acoustic sensors; and increasing of optical absorption and consequent decrease in receiving power in optical sensors [5,6,7,8,9,10,11].

Additional problems associated with biofouling emerge in the case of marine optical sensors where high optical transparency is required. Before deployment, calibrations are needed for the good operation of most monitoring instruments. Long-term monitoring can result in biological attachments in the sensor surface that may require hard mechanical cleaning methods, such as high-pressure water jets or brushes, or chemical methods. These methods can modify the initial status of the sensitive sensor area, and consequently, make the comparison of the metrological response of the sensor before and after the deployment difficult. Moreover, bad cleaning will have the same result [12].

Some commercial anti-fouling techniques are available and applied to oceanographic instruments. The most known are the biocide generation systems based on copper corrosion mechanisms or TBT (tributyl tin) leaching [13]. Even though the use of TBT leaching has proven to be efficient, it presents adverse environmental effects, so was banned for antifouling paints and later for ships’ hulls [14]. Thus, protections based on TBT leaching cannot be considered a solution for biofouling protection.

Nevertheless, copper biocide properties can still be used to protect the monitoring instruments. This protection, usually called “copper shutter”, uses a copper housing with a mechanism with a motor driver and a shutter that only opens when taking measurements. The sensor is kept inside the copper shutter that remains closed while the sensor is not taking measurements, in the darkness, and allows biocide concentration to increase, thus preventing biofilm formation [15].

Another mechanical anti-fouling technique, and the most common in use for commercial instruments, is the use of wipers or scrapers in the sensing area [16,17]. This technique is effective if the wipers or scrapers are in proper condition and the sensor head is suited for the cleaning process. The major disadvantage of these mechanical systems is the complexity of the watertight needs of the mechanical moving parts, and it must be implemented by the early stages of the sensor development and cannot be adapted to old ones. Additionally, the motor used to move the mechanical gears needs power from the batteries, which decreases the operation time of the instrument.

New technologies have been presented in the recent literature to reduce the impact of biofilm, as also ways of controlling its presence on the surface of sensors. Different active techniques based on ultraviolet [18,19,20], laser [21,22], direct electrification [23,24], bubbles [25], or acoustics [26,27,28] were demonstrated to be effective but presented high energetic requirements for continuous and long-term monitoring.

Another active technique is based on the use of chlorine, both by in situ production by salty water electrolysis or by bleach injection [29,30]. Primarily used in industrial applications, chlorine techniques have recently migrated to oceanographic instruments as is the case of Wet Labs/Sea-Bird WQM (the device uses a reservoir for the chlorine solution and a pump to inject it into the surface of the sensor) or electrolysis chlorination systems in monitoring stations [31]. Other techniques based on chlorine production that focused on protecting only the sensing area of the instrument, have been presented [32]. These techniques have less energetic needs and are more compliant with long-term monitoring power requirements.

Finally, nano-coatings of the glass surface, coatings of the sensing area of the sensors or bio-mimetics shapes and materials are emerging techniques that have shown their potential mostly in laboratory experiments [33,34,35,36,37].

Even though biofouling is far from being a solved problem, many techniques have been presented in the last decades to extend the deployment time of these instruments. However, many of them are also still restricted to laboratory conditions and sea truth validation of their reliability is still needed.

We have been developing several cost-effective oceanographic optical instruments for in situ continuous monitoring [38,39,40,41]. In these instruments, the structural housing is typically built using 3D printed material such as polylactic acid (PLA) or acrylonitrile butadiene styrene (ABS), which is less robust and more susceptible to biofouling when compared to the material used in typical commercial sensors (e.g., titanium or sapphire glass). For this reason, biofouling is a major concern, particularly when these cost-effective sensors are intended to match the performance of their commercial peers. With this problem in mind, we tested the effectiveness of six anti-biofouling techniques applied to a turbidity optical sensor under sea truth conditions.

## 2. Materials and Methods

### 2.1. Design of the Probes

Sensors that make use of optical techniques are usually more susceptible to biofouling, decreasing their sensitivity and reliable lifetime. Even a very thin biofilm on the surface of the optical regions may produce interference in the measurements due to absorption and/or total or partial light scatter. If we consider that biofouling starts its formation the moment that the instrument is submerged, optical sensors can have a very short time of reliable operation.

We presented an optical sensor to measure turbidity and suspended particulate matter using transmitted light, backscattering, and nephelometric light techniques [39]. For the transmitted detection, both fouling on the surface of the optical transducers and the existence of macro-fouling in the sensor housing can block the passage of light and increase the turbidity values measured by the sensor. However, for the backscattering and nephelometric techniques, the decrease in the optical signal results in lower turbidity values, but macro-fouling can also generate undesired scattering that increases the turbidity measurements. This factor presents additional challenges when analyzing the biofouling interference for the backscattering and nephelometric techniques.

To reduce the complexity of the biofouling effects in the measurements, we narrowed the problem to the use of just the transmitted light technique, using infrared wavelength, as used by many optical oceanographic sensors.

Six probes were built using different anti-biofouling techniques or housing properties. The probes were designed in a tube shape with an optical measuring area in its inside. This shape was chosen to create a sensing area with low fluid flow and turbulence so that some of the techniques (localized chlorine production and copper biocide) can be more effective rather than in an open design, unprotected from the normal stream flow.

Each one of the probes uses an infrared 980 nm light emitting diode (LED) and one infrared 980 nm phototransistor that measures light at 180° to the LED (transmitted light detection technique). Figure 1 shows the 3D design and transducers’ schematic of the probes.

The sensor housing presented in the image above was used as standard and was slightly changed to adapt to the different anti-biofouling techniques tested. These techniques can be divided into four different groups: structural material, copper biocide, transparent coating, and chlorine production.

#### 2.1.1. Structural Material

The characteristics of the surface which houses the sensor play a major role in the attachment of biofilm. Smooth faces are less susceptible to attaching biological organisms when compared to rough surfaces (in fact, these surfaces are sought to design artificial reefs for example [42]). In addition, materials such as inox or titanium are known to be less propitious to biological film formation when compared to plastic ones.

However, the most effective materials against biofouling are also more expensive. Since the main objective of this work is to test the application of anti-biofouling techniques in low-cost instruments, we tested the two cheapest and wider in-use 3D printing materials: PLA and ABS.

It is important to notice that the PLA and ABS probes are not anti-biofouling techniques but rather the use of different structural materials. In both probes, the optical transducer surfaces are in contact with the water without any protection. However, both ABS and PLA materials are often catalyzed with organotin which can leach out and act as antifoulants [43,44].

The performance of ABS and PLA probes in the field is not expected to differ significantly, which would allow their use as control samples to be compared with the other techniques.

The PLA and ABS probes are shown in Figure 2 as probes 4 and 1, respectively.

#### 2.1.2. Copper Biocide

This technique, PLA with copper filament (FlashForge PLA Copper 1.75 mm), is also based on 3D printed material for the structural housing. As before, the surfaces of the optical transducers are in contact with water. However, as discussed in Section 1, copper has biocide properties that prevent the formation of biofilms. Therefore, once the transducers are comprised inside the tube shape, it is expected that this technique extends the reliable operation time when compared to PLA and ABS.

The advantage of the use of this technique is that it is easy to apply (just the use of different materials in the 3D printer) and it is still a cost-effective material. The main disadvantages are that the copper protection has a limited time since it loses its biocide properties over time and, as discussed before, the biocide is not environmentally friendly.

The PLA with copper filament probe is presented in Figure 2 as probe 2.

#### 2.1.3. Transparent Coatings

Contrasting with the previous techniques, the coating techniques are used to protect the transducer surfaces, which are no longer in contact with water. In this technique, the formation of biofilm will happen on the surface of the coating that is supposed to have better anti-biofouling properties. However, optical sensors present an enhanced challenge since the coatings must ensure the passage of light. Attending to this need, we selected two easy-to-apply transparent materials: transparent epoxy resin (HB EPOSURF2—HBQUIMICA) and polydimethylsiloxane (PDMS—Sylgard^®^ 184 Silicone Elastomer Kit). Both probes were built in ABS and the optical transducers were coated with transparent materials. The idea of using both PDMS and epoxy as anti-biofouling coatings has been presented before [45,46,47,48,49].

Epoxy resin is an affordable and widely used material that is easy to apply. The resin takes about 7 days to cure before it is ready for immersion. If the coating is submerged earlier, there is the possibility of its surface becoming whitish.

The PDMS is an affordable material with wide use in MEMs fabrication and, although it needs dedicated machinery to cure, it is also easy to apply.

The main disadvantage of these techniques is that even though the materials are transparent, they still cause light attenuation, thus causing changes in the measurements delivered by the sensors. If the sensor is built from scratch, this attenuation can be compensated with electronic instrumentation, however, if the sensor is already built and the coating is applied afterward, new calibrations are required. It is important to notice that as for the structural materials, the effectiveness of the coatings depends on their anti-biofouling properties; in the case of PDMS the release of organotin and for the epoxy other toxic compounds.

The epoxy and PDMS coating probes are presented in Figure 2 as probes 5 and 3, respectively.

#### 2.1.4. Chlorine Production

Chlorine is used worldwide for water disinfection. Here, we use the electrolysis of salty water to produce it. In this approach, the electrodes are usually based on platinum (Pt), dimensionally stable anodes (DSAs) of a Ti-support coated by noble metal oxides such as ruthenium, iridium, tantalum, zirconium, and doped diamond electrodes. Although this technique is very effective to produce chlorine biocide, the opacity of the electrodes has always been an obstacle to its application in optical instruments.

We presented a new approach to chlorine biocide production combining the advantageous features of the platinum material with transparent conductive oxides [50,51]. In this technique, the anode electrode, where the chlorine is produced, is transparent and can be directly applied above the optical transducers (similar to the transparent coating techniques). This electrode is constituted by Pt nanoparticles coated with transparent conductive fluorine tin oxide (FTO) thin film supported on the glass substrate. A stainless-steel electrode was used as a cathode for long durability in water without oxidation. Considering the laboratory experiments of Pinto et al. [51], the electric power for the chlorine production in this probe was designed to be 1050 µW, in a 2.5 cm^2^ area. The production was set to be on all the time and only turns off when the sensor is collecting data. With this configuration, the sensor is achieving a production rate of approximately 4 mg of chlorine per hour.

The main disadvantage of this technique is the need for electric power which reduces the operation time of the sensors. However, it allows controlling the amount of chlorine biocide to the minimum required to avoid biofouling, maintaining the sensor area cleaned without negative impact on the marine environment. This technique is only suited for marine environments since it does not work in freshwater.

The chlorine production sensor probe is presented in Figure 2 as probe 6.

The six probes were assembled on one board with a minimum distance of 1 cm from each other. The probes based on biocides production are not expected to affect other probes through the spread of the biocides due to the confined shape of the probes. As demonstrated by Pinto et al. [51], chlorine production is a local process restricted to the surface of the glass substrate. On the other hand, the structural housing of the PLA with copper filament probe was all built with this material, including the outside walls of the probe, which means that the release of copper biocide is not totally confined. Still, in open waters, the release of copper is not expected to be high enough to influence the biofouling protection of the other probes, and if it happens, it will only affect the outside walls and not the measuring areas of the probes.

### 2.2. In-Lab Calibration of the Probes

Before deployment at sea, the probes were calibrated to correlate the electrical output to the corresponding value of turbidity. As in previous work from the authors [41], a standard formazine solution was used to calibrate the probes into nephelometric turbidity units (NTU).

An initial solution of 4000 NTU was diluted in distilled water to obtain solutions with lower turbidity values. The ratio proportion of the initial and final solutions is given by the following equation:(1)$$Dilution_{factor}=\frac{volume_{total}}{volume_{initial\_sample}}=\frac{volume_{dilluted\_water}+volume_{initial\_sample}}{volume_{initial\_sample}}$$

The probes were submerged in a container with different turbidity solutions. Different dilution factors were used to calibrate the sensors for 1000, 500, 250, 125, 62, 31, 15, 7.5, and 3.8 NTU.

Figure 3 shows the calibration of each probe with the turbidity solutions. During the development of the instrument, the electric gain of each technique was set so that for diluted water (low turbidity values) the corresponding outputs were similar. For high turbidity values, the sensitivity of the epoxy, PDMS, and chlorine techniques was lower compared to PLA, ABS, and copper (different slopes in the curves). This is a result of the coating of the optical transducers for the first group, while for the second the LED and photodetector are in contact with the medium. Other differences in the values obtained from the different techniques are related to the alignment of the optical transducers, which are assembled by hand.

Each one of the curves was used to convert the electrical output of the probes to turbidity values during the in situ test. After the experiment and cleaning of the sensors, a new calibration was conducted to check if the probes kept the same output as before the experiment, or if there was a decrease in the signal resulting from the biofouling.

### 2.3. In Situ Deployment

The probes were deployed, from 23 May to 9 July 2022, in the dock of the Marine Biological station of the University of Vigo, located at Toralla Island, Ria of Vigo, Spain (42°12′07.1″ N 8°47′54.6″ W). The Ria of Vigo is a highly productive coastal ecosystem (primary production rates of up to 3 g cm^−2^ d^−1^) [52], characterized by clean waters and low sedimentary dynamics which makes it ideal for shellfish farming, and consequently for biofouling.

The instrument was moored attached to a post of the floating dock, at 1.5 m depth from the surface, with the opening of the probes faced to the sea bottom (less influence of the daylight in the measurements). This location has a depth variability of 5 to 7 m during tides.

The probes were connected by an electric cable, that shares power and communications, to a data logger that recorded the measurements of the six anti-biofouling techniques with a sample rate of one reading per hour. The whole system was supplied by the electrical grid available on the dock.

### 2.4. Cleaning Process

Two days before the end of the in situ experiment, the probes were taken out of the water for soft cleaning. This cleaning process, referred to as in situ cleaning hereafter, consisted of the removal of the macro biofouling attached to the sensors and the cleaning of the inside of the probes with fresh water and a soft cloth. After the in situ cleaning, the probes were deployed again for two more days to confirm the results.

After the in situ experiment, the sensors were taken to the laboratory for hard cleaning. The probes were submerged for five days in a water tank with a high concentration of chlorine. This process killed and detached the algae from the instrument, leaving just the cirripeds, which were then removed. Finally, and with the probes clean of macro biofouling, the inside of the probes was cleaned with fresh water and a soft cloth.

## 3. Results

The efficacy of the developed techniques was evaluated by three different metrics: the analysis of the turbidity measurements of the probes during the in situ experiment; visual inspections of the probes; and the comparison of calibrations before and after the in situ test.

### 3.1. In Situ Experiment

Figure 4 shows the measured turbidity of each probe during the field experiment. The experiment started on the 23 May 2022 with all the probes measuring turbidity values of 50–60 NTU. It is important to notice that turbidity is not expected to change significantly in this area, so the output of the probes should remain constant without biofouling interference. However, since biofouling is certain, during the following weeks, the different anti-biofouling techniques presented a drift in the measurements at different stages of the test. On the 7 July, the probes were taken out of the water for the in situ cleaning (red vertical line on the graphics of Figure 4) and deployed again for two more days.

The ABS and PLA, which were not expected to provide biofouling protection, took about five days to present a significant drift in the signal. The turbidity measurement delivered by the PLA probe increased during the experiment, a signal of biofouling interference. For the ABS, turbidity increased till the 28 July and then the values decreased to about 100 NTU, which suggests that macro-fouling attached to the sensor, for example, algae, blocked the passage of light in the sensing area of the sensor (turbidity increases). Measurement gets back to normal values once algae are detached (in this case slightly above the values at the beginning of the experiment because of micro-biofouling that is still present on the surface of the optical transducers). Previous field tests conducted with similar optical instruments [38,39,40,41] lead us to interpret this high variability in turbidity measurements as derived from the attachment and unattachment of macro-fouling. However, we cannot exclude the possibility that in some situations the organotin compounds released by the materials could also produce an antifouling effect.

The PLA with copper filament presents a significant drift from the 8 June. Compared with the ABS and PLA probes, which also have optical transducers in contact with water, the releasing of copper biocide into the water seems to increase the operation time of the sensor without interferences of biofouling. While this technique was also affected by biofouling, the measurements are more constant when compared to the other material techniques.

The probes with epoxy and PDMS coatings did not demonstrate effectiveness against biofouling. The epoxy measurements started with values of 56 NTU and in the first 3 days of the experiment, it increased to values around 100 NTU. From the beginning of the experiment to the 29 June, this probe produced measurements of around 100 NTU but showed a high variability (about 40 NTU in amplitude). From this day to the in situ cleaning, the sensor registered an increase in the turbidity values, probably caused by macro fouling.

The PDMS probe produced very similar results. In the first 3 days of the experiments, the turbidity measurements increased from 50 NTU to 150 NTU, and this value slowly increased to around 200 NTU on the 30 June. As for the epoxy technique, the recorded turbidity measurements increased the days before in situ cleaning.

The last technique, chlorine production, produced the best results among the six probes. Since the first day of the experiment, the probe presented values of 50 NTU with a maximum 5 NTU variation in its measurements. On the 5 July, an increase in the turbidity values was observed, probably associated with the beginning of the drift. The measurements produced by this probe are the expected without biofouling interference.

On 7 July, the probes were cleaned and deployed again on the water to check if the in situ cleaning process was sufficient to eliminate the biofouling interference. Good results were obtained for the PLA, ABS, and chlorine probes since the turbidity measurements after the cleaning dropped to similar values at the beginning of the experiment.

For the epoxy and PDMS techniques, the cleaning also produced positive effects decreasing the recorded turbidity values. However, these values were still higher than the 50 NTU at the beginning of the tests, which indicates that the micro-biofouling was not eliminated.

Finally, the cleaning did not produce any effect on the copper probe and the turbidity values of the measurements kept increasing till the end of the experiment.

### 3.2. Visual Inspections

The objective of the visual inspections was to provide qualitative comparisons between the probes with respect to the quantity of biofouling at different stages of the experiment (mostly macro-biofouling).

Photographs of each probe were taken before the in situ test, before and after the in situ cleaning on the 7 July and after the hard cleaning when the experiment ended. Figure 5 shows the state of each probe at the four different moments.

When the instrument was taken out of the water on the 7 July for the in situ cleaning, the external housing of all probes was covered with macro-fouling, namely algae, barnacles, and some small mussels. However, there was a visible difference in the amount of macro-biofouling between the outside and the inside (sensing area) of the probes. The inside of the probes had fewer traces of biofouling, even for the PLA and ABS substrates, probably because the instrument was deployed facing the seafloor and the inside of the probes was protected from direct sunlight, which contributes to the biological growth.

The in situ cleaning procedure removed the algae, and the interior of the probes was cleaned with fresh water and a soft cloth. The observation of the probes after the in situ cleaning (third row of Figure 5) showed that while the external housing had some barnacles, the sensing area was clean.

At the end of the field tests, the instrument was submerged for five days in water with high chlorine concentration to remove the macro-biofouling and the micro-biofouling in the surfaces. After this process, the barnacles and mussels attached to the sensors weakened and were removed by hand. Some of them, however, had to be removed using a chisel. The inside of the probes was cleaned with fresh water and a soft cloth. The state of the probes after this cleaning procedure is shown in Figure 5 (fourth row).

The comparison of the photographs before the field tests and after the hard cleaning shows that the external housing of the probes was damaged (as expected given they were built with cheap materials). However, the sensing areas were clean and in good condition. After the hard cleaning, all the probes had similar inside conditions as before the experiment, except for the PDMS coating, which was not as transparent as before, presenting a dark blur.

It is important to notice that both coating techniques (PDMS and epoxy) and the chlorine production probe did not have any trace of macro-fouling in their surfaces at any stage of the visual inspections. Additionally, the copper technique was the probe that visually had fewer biofouling traces on its inside. However, during the hard cleaning, we noticed that a barnacle was growing on the surface of the LED, which was probably the reason why the in situ cleaning did not result in lower turbidity values (see data presented in the previous sub-section).

### 3.3. Calibration Signal Loss after the Field Experiment

After the in situ test and the hard cleaning, the sensors were calibrated again to check if the probes kept the same output as before the field experiment. Figure 6 shows the relative sensor performance after the second calibration in comparison to the first calibration presented in Figure 3. A value of 100% represents that the sensor kept the same performance as before.

The PLA, chlorine, and copper probes presented the lowest performance loss from the first to the second calibration. The turbidity measurements using the formazine solutions in the second calibration were practically the same as for the first one. The ABS probe presented an average relative performance of 96%, which is lower than the previous ones, but still acceptable.

The coating materials are the techniques that presented lower performances, displaying the PDMS with the worst results with a relative performance of only 59%. This result is consistent with the visual inspections after the hard cleaning when it was noticed that this coating was with a dark blur on its surface. Even with the cleaning process, these techniques did not present similar calibration results as before the field test.

## 4. Discussion

Considering the three metrics used to evaluate the operation of the six anti-biofouling techniques, we conclude that chlorine production outperformed the other tested techniques. The measurements carried out during the field test presented a signal without drift and the interference of biofouling was only detected after 41 days of testing, during the visual inspections the inside of the probe always appeared clean, and the calibration of the probe did not show changes after the deployment at sea.

The PLA with copper filament also produced satisfactory results in the field tests. Compared with the other techniques (except the chlorine), the drift originated by biofouling interference was detected later, and the probe provided 15 days of reliable data. This was the only technique where the in situ cleaning did not have positive effects, possibly due to the growth of barnacles on the sensors LED surface that was only detected during the hard cleaning. This is one disadvantage of this technique, since the copper biocide, although satisfactorily protected the probe housing (during the visual inspections this technique was highlighted as the less susceptible to biofouling growth on its surface), failed to protect the transducer surface.

The results for the other four probes did not provide insights into good protection against biofouling. During the field tests, the epoxy and PDMS presented biofouling interference from the day of deployment. However, both stabilized on 100 NTU and 150 NTU, respectively, till almost the end of the experiment. Even the in situ cleaning was not sufficient to obtain turbidity measurements similar to those of the first deployment day. In the second calibration, these techniques presented the worst results, with the PDMS having a significant signal loss resulting from the blur of its coatings. All these factors suggest that these coatings are very susceptible to micro-biofouling, even to an irreversible state.

Finally, the PLA and ABS substrates, which were used for comparison with the other techniques and were not as effective anti-biofouling protections, presented the worst results during the field tests. The PLA probe produced satisfactory measurements till the 5th of June, but after that day the drift in the measures increased significantly. For the ABS probe, the drift was visible since the day of deployment, and contrary to the epoxy and PDMS that suffered mostly from micro-biofouling, macro-biofouling interference was constant throughout the experiment. On the positive side, the in situ cleaning provided good results and the hard cleaning left the probes in the same conditions as before the experiment.

The results obtained in this investigation are consistent with those theoretically expected. Chlorine is a strong oxidant that destroys primary biofilms and microbes, and propagules of macro-fouling. The anti-fouling efficiency against the primary film is crucial for a long-term operation without biofouling interference in the measurements. The other techniques tested, based on the release of organotin and toxic compounds into the water, may also be harmful to macro-fouling organisms but are not as effective as chlorine biocide against biofilm formation. The anti-fouling efficiency of copper biocide has been previously validated, as shown in the scientific literature and, consequently, satisfactory results would be expected from this experiment.

## 5. Conclusions

In this work, we present the design and field experiment of different anti-biofouling techniques to be applied to optical instruments. Six different probes were built based on different materials: the housing of the probes in PLA and ABS; copper biocide due to the housing of the probe in PLA with copper filament; chlorine biocide production by the electrolysis of seawater; and transparent coatings using epoxy and PDMS.

All the probes were calibrated before and after the field experiment using formazine solutions. The system was deployed in a coastal area with low sedimentary dynamics from the 23 May 2022 to the 9 June 2022 with the probes taking measurements with a sample rate of one hour. Different procedures for probe cleaning were used during and after the field experiment.

The effectiveness of the tested techniques was evaluated by three different metrics: the monitoring results during the field test, visual inspections of the probes during and after the experiment, and the relative signal loss of the calibrations before and after the test.

The chlorine production technique outperformed the other ones, producing reliable measurements along the experiment, and the results show effective protection against micro and macro-biofouling. The results for the copper biocide were not as good as for the chlorine but also demonstrated to be an effective technique. On the opposite, the PDMS, epoxy, PLA, and ABS did not yield satisfactory results for biofouling protection.

Future work should now focus on the chlorine production technique. For this experiment, the chlorine probe was designed to produce seawater electrolysis using 1050 µW of electric power, but similar results might be achieved with lower power. Since power efficiency is a concern for long-term monitoring, the optimal power to protect against biofouling must be addressed. In addition, the copper technique should not be discarded, and other structural housings based on copper materials can be tested.

Biofouling is still a problem that affects the performance of optical instruments and has been obstructing long-term monitoring studies with fully automated systems, without the need for pauses in the tests for cleaning, new calibrations, or other maintenance purposes. The scientific community has been focusing attention on this problem and new and innovative technologies have been emerging. However, most of these technologies are still confined to the laboratories and have not been truly applied in real sensors and tested in the field. We believe that experiments like the one we presented must be replicated to fully understand the potential of these emerging technologies, which can only be accomplished when applied to devices measuring physical or biological water parameters and tested in the field.

## Figures and Tables

**Figure 1 sensors-23-00605-f001:**
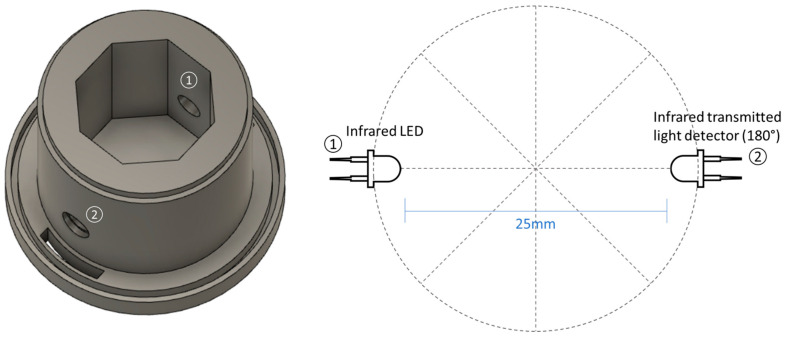
Three-dimensional drawing of the probe comprising the optical transducers (**left image**) and scheme of the positions of the LED (1) and the transmitted light detector (2) (**right image**).

**Figure 2 sensors-23-00605-f002:**
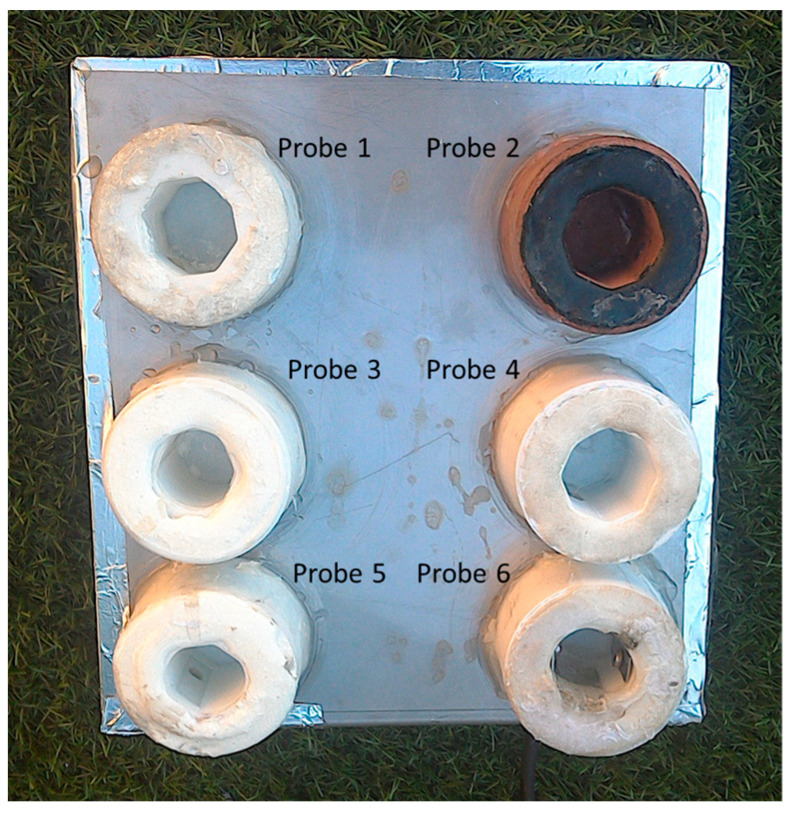
Anti-biofouling techniques probes. Probe 1: ABS material; Probe 2: PLA with copper filament; Probe 3: PDMS coating; Probe 4: PLA material; Probe 5: epoxy coating; Probe 6: chlorine production.

**Figure 3 sensors-23-00605-f003:**
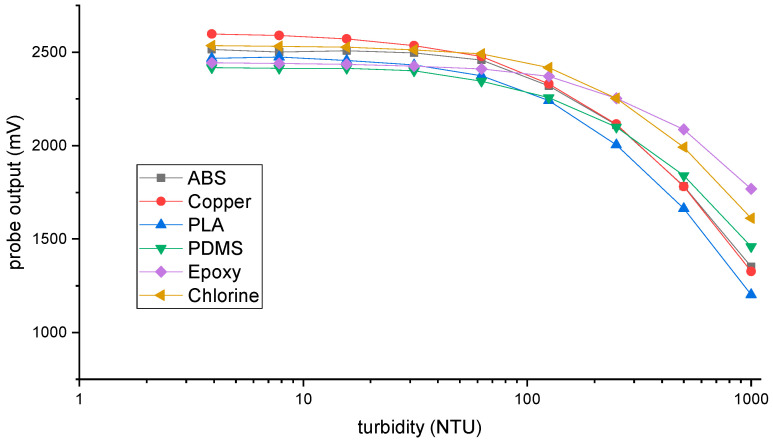
Calibration of the six probes with formazine. The electrical output of each detector is correlated to the different turbidity solutions.

**Figure 4 sensors-23-00605-f004:**
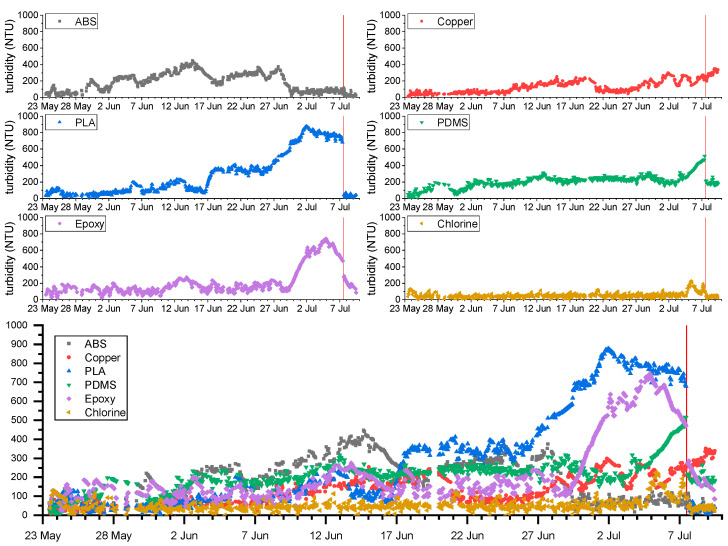
Turbidity measurements of the six anti-biofouling techniques during the in situ experiment. The vertical red line marks the date when the probes were taken out of the water for the in situ cleaning.

**Figure 5 sensors-23-00605-f005:**
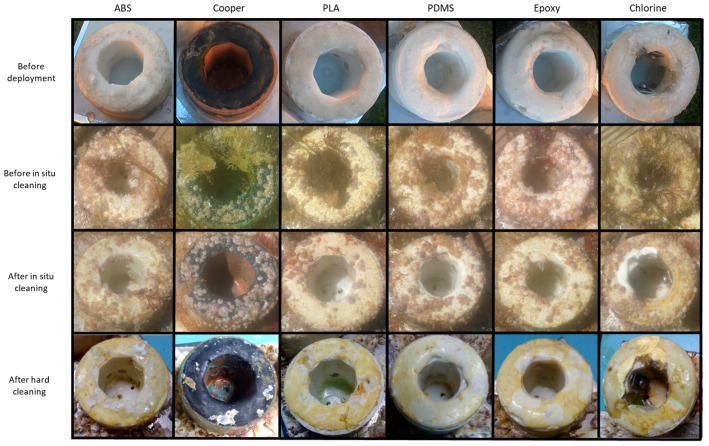
Photographs of the six probes at different stages of the experiment: before deployment, after 46 days of deployment, after the in situ cleaning on the 7 July, and after the hard cleaning at the end of the field test.

**Figure 6 sensors-23-00605-f006:**
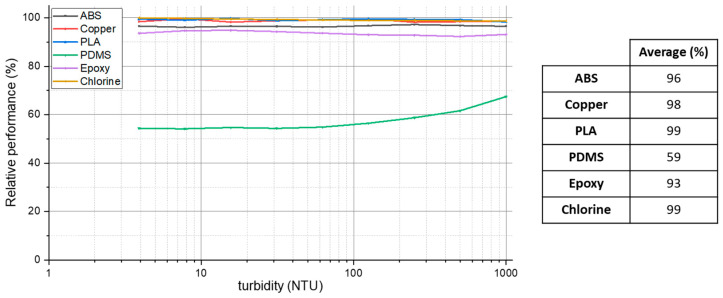
Relative performance between the first and the second calibration after the in situ experiment for the six anti-biofouling probes.

## Data Availability

Data sharing not applicable.

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
