# Peer review of "Design and In Situ Validation of Low-Cost and Easy to Apply Anti-Biofouling Techniques for Oceanographic Continuous Monitoring with Optical Instruments"

_sensors, 2023, doi:10.3390/s23020605_

Round 1

Reviewer 1 Report

The authors built six different optical probes based on different anti-biofouling materials or techniques to investigate their anti-biofouling efficiency. The probes were carefully designed and field experiments were carried out. Results show that the chlorine production technique has the best performance, the copper technique had lower performance but still useful, while techniques based on transparent coatings, epoxy and PDMS, could not prevent the biofilm formation. The investigation proposal and the results are of guidance to the choice of anti-biofouling techniques for optical sensors. This work could be published after careful revision. The followings are questions for revision:

1.     From Figure 2, all the probes were arranged on one board and they are not far apart. Do they have influence on each other? For example, the chlorine produced from probe 6 may spread to other parts. How about the influence?

2.     The biofouling could randomly appear anywhere on the probes. If it happens to take place on the surface of the LED or the light detector, the influence to the measurement results would be more obvious, which may also induce difference between probes, and probes with the same anti-biofouling techniques may show different results.

3.     Why chlorine production outperformed the other tested techniques. Authors should give scientific explanation. 

4.  Anti-biofouling technique based on transparent coatings is important for optical sensors.  What is the logic and reasoning to choose PDMS, epoxy, PLA and ABS for study? since these materials are not suitable for marine application.

Reviewer 2 Report

sensors-2089971

This is a nice piece of work that should be a good contribution once you have attended to details

The biggest detail is Chlorine is a strong oxidant that kills all living biology and removes biological films including adsorbed molecules 

PLA is catalized with organotins at a level to be toxic to macrofoulers even without the addition of copper. I don't know about ABS

Figure 2 it woudl be good to expand the first parts of the figure to show what happens up to about June 8 with the lines on the same figure. so you can actually copare. 

Need to give the amount of chlorine produced per some unit time.

line number.       Comment

Introduction

You could remove many  the from the text in a variety of sentences. Would save space.

35 biofouling is the accumulation of biological material from molecules to metazoans on surfaces.

38 some biofouling is virtually instantaneous.  See Clare et al. Invertebrate reproduction and behavior 1992

  Primary film is biofouling.

See Xiaobo Liu, Jinlong-Yang, Daniel Rittschof, James S. Maki, Jidong Gu 2022. Redirecting antibiofouling innovations from sustainable horizons. 2022. Trends in Ecology and Evolution. https://doi.org/10.1016/j.tree.2022.02.009

The rest of the introduction is fine

Materials and methods

Since you are doing short-term the following comments are important for certain kinds of biofouling-  since biofilms grow on organotin protected surfaces they are not important to others.

175—ABS and PLA are not the same using either as a control sample needs to be demonstrated.  PLA is often catalyzed with organotins which can leach out and act as antifoulants though you say organotins are not used for antibiofouling hey are still used as catalysts and their amount is often 250 micrograms per gram polymer 4 ug of organotin per square cm release kills everything—organtins aren’t very solubile they move to the surface and kill at the surface

198—what is the caralyst in Silgard 184?

222-236. Chlorine is a strong oxidant—it oxidizes primary film—how much is produced per unit time?

274. highly productive waters are good for shellfish farming how productive are the waters?

Results-  Only chlorine is an oxidant that destroys primary film and microbes and propagule of macrofoulers. Pla contains organotin.  Curing epoxy is releasing toxic copmpounds.  Silgard contains organotins

Organotins do not work on algae.

313 PLA had growth until organotins bloomed to surface

Abs shows the same trent fouling then unfouling

PLA looks like copper until the 6th of June

Reevaluate your results with the knowledge that many of the coatings release molecuels toxic to macrofoulers in the short term (2 months)

With my comments about toxic components your results make sense.

Discussion

You need to talk about chlorine being a strong oxidant and that oxidizing primary film is important.

441—your data in figure  do not match your discussion especially PLA from 28 May to 6 June. Looks a whole lot like copper PLA

Round 2

Reviewer 2 Report

Design and in situ validation of low-cost and easy to apply 3 anti-biofouling techniques for long-

term oceanographic 4 monitoring with optical

I am concerned that you are using short term (48 days) and the title is long-term.

line.                                    Comment

17                                      delete- that has been

18                delete- The in the first full sentence

22.               delete - different materials namely

Continue trying to remove as many unneeded words as possible.

108              primordially used should be primarily

125.             3D printed not printing

128              delete-even

187            delete- from each other

227            epoxy other toxic compounds

231.           delete- a known compound

Figure 4. Combine  copper   PLA into one panel—PLA is organotin Copper is Organotin+ copper
